# New Policy and Implementation of Municipal Solid Waste Classification in Shanghai, China

**DOI:** 10.3390/ijerph16173099

**Published:** 2019-08-26

**Authors:** Ming-Hui Zhou, Shui-Long Shen, Ye-Shuang Xu, An-Nan Zhou

**Affiliations:** 1State Key Laboratory of Ocean Engineering, Department of Civil Engineering, School of Naval Architecture, Ocean, and Civil Engineering, Shanghai Jiao Tong University, Shanghai 200240, China; 2Key Laboratory of Intelligent Manufacturing Technology (Shantou University), Ministry of Education, and Department of Civil and Environmental Engineering, College of Engineering, Shantou University, Shantou 515063, China; 3Civil and Infrastructure Engineering Discipline, School of Engineering, Royal Melbourne Institute of Technology (RMIT), Victoria 3001, Australia

**Keywords:** MSW classification, policy, implementation, Shanghai

## Abstract

To promote sustainable development, the Chinese government launched a new municipal solid waste (MSW) classification strategy in 2017. Shanghai was selected as one of the first pilot cities for MSW classification. The Shanghai municipal government first established the new MSW classification policy in 2017. The Shanghai Municipal Solid Waste Management Regulation was published in 2019 and came into effect on 1 July 2019. This short communication reports on Shanghai’s new MSW classification policy and its implementation. The main content and measures adopted by Shanghai’s government to ensure the effective implementation of the new MSW classification policy are introduced. Besides, a SWOT (i.e., strengths, weaknesses, opportunities, and threats) analysis on the present policy and measures is conducted, and based on the results, some discussions and suggestions regarding the implementation of MSW classification in Shanghai and the whole of China are presented.

## 1. Introduction

Owing to rapid growth in urbanization and economic development, municipal solid waste (MSW) has become a serious problem in China, especially in mega cities [1,2]. MSW refers to solid waste generated by households and commercial and governmental enterprises in daily life including waste food, melon peel, fruit cores, clothing, waste metal and glass, etc. [3,4]. According to the latest statistical data [5], 202 major cities in China produced a total of 202 million tons of MSW in 2017. Many cities are in a “waste siege” dilemma, where cities are surrounded by piled waste in suburban or rural areas [1,6]. A large amount of MSW becomes a hazard, which not only affects the environment but also endangers human health [7,8,9]. MSW classification might be an effective way to deal with the waste dilemma [3,10,11].

MSW classification includes the classified throwing, collection, transportation, and disposal of MSW based on certain standards [4]. Historically, both central and local governments in China tried several times to promote an MSW classification policy, but the effective implementation of MSW classification was not possible owing to multiple factors, such as ill-willingness of people, insufficient technology and infrastructure, poor coordination among different departments of the government, and lax laws and regulations [11,12,13,14,15]. In recent years, China changed its development strategy from one based on rapid development to another with a focus on environmental protection. The significance of MSW classification is thus reiterated. In the 2017 Government Report, Chinese premier Keqiang Li announced that China would promote MSW classification [16]. Some key cities were then selected as pilot cities to implement the new MSW classification policy by the National Development and Reform Commission (NDRC) and the Ministry of Housing and Urban-rural Development (MHUD) of China in 2017. Shanghai, as one of the most developed cities in China, is one of the first pilot cities for MSW classification [17,18]. 

Shanghai produces nearly 9 million tons of MSW per year, and this amount is continuingly increasing [5,19]. Shanghai’s new MSW classification policy captured people’s attention not only in Shanghai but the whole of China. If MSW classification in Shanghai is successful, Shanghai’s MSW classification policy will be an exemplary model and will serve as a benchmark to other cities [20]. To adhere to the new national development strategy, various regulations have been compiled and measures have been adopted by both the municipal and district governments of Shanghai to promote the new MSW classification policy since 2017 [21,22]. On 1 July 2019, the Shanghai Municipal Solid Waste Management Regulation [4]—the first local mandatory regulation for MSW classification in China—came into force, which indicated that MSW classification has been incorporated into the legal framework in Shanghai. Both government and citizens of Shanghai are now contributing to MSW classification, aiming to build a complete, world-class MSW classification system by the end of 2020 [20,21,22,23].

This article reports on Shanghai’s new MSW classification policy and its implementation. First, the main contents of the new MSW classification policy of Shanghai are summarized. Then, some measures adopted by the government to ensure the effective implementation of the policy are introduced. Finally, discussions and suggestions regarding how to improve and promote the MSW classification policy in China are presented. 

## 2. Policy and Measures

### 2.1. New Policy

According to the Shanghai Municipal Solid Waste Management Regulation [4], Shanghai classifies MSW into four categories, namely recyclable, hazardous, wet, and dry waste, defined as follows: (1) Recyclable waste refers to the waste that is suitable for recycling, for instance, used paper, plastic, glass, metal, and fabric. (2) Hazardous waste refers to the waste that might cause direct or potential harm to the human health or natural environment, such as waste batteries, lamps, drugs, paints, and pesticides. (3) Wet waste refers to the perishable biomass waste such as leftovers, expired food, melon peel, fruit core, and dead flowers and plants. (4) Dry waste refers to any waste other than recyclable, hazardous, and wet waste. New waste collection points have been deployed in almost every corner of Shanghai [20]. Figure 1 shows a typical waste collection point with four waste collection bins in Shanghai. It is required that hazardous, recyclable, wet, and dry waste be thrown into red, blue, brown, and black bins, respectively. Citizens are required to classify waste first and then throw it into the right bins. Unclassified waste will not be collected. Furthermore, citizens are required to throw the waste in prescribed places and times. No waste collection outside of the prescribed time slots and places is permitted [4].

The classified waste will then be collected by qualified enterprises and transported to different sites for appropriate disposal [4]. Recyclable waste will be recycled by recycling enterprises for resource utilization. Hazardous waste will be transported to the waste treatment plants for innocuous treatment through high-temperature, chemical, and other treatment processes. Wet waste will be used to produce biogas or utilized as natural fertilizer after complex chemical and biological treatments in waste treatment plants. Dry waste will be incinerated in waste incineration plants to generate electricity or dumped in landfills. 

Different types of waste will be transported by different kinds of specified waste transportation vehicles [4]. Thousands of waste transportation vehicles have been deployed to collect and transport the waste in a timely manner [20,24]. Figure 2 presents four typical waste transportation vehicles in Shanghai. Currently, ten new waste treatment plants are under construction in Shanghai [25]. Once these plants are completed, the total daily disposal capacity of waste in Shanghai is expected to increase to 32,800 tons per day by the end of 2020 [20]. 

Efforts are also made to decrease the waste production from the source. Governmental agencies and institutes are required to use products and equipment compatible with the environment [4,22]. Recyclable paper is recommended, and disposable office supplies are discouraged in the daily operation of the government. All enterprises, companies, and factories in Shanghai are encouraged to give priority to detachable, recyclable, and non-toxic materials and designs, and produce environment-friendly recyclable products. Restaurants, shops, and hotels are not allowed to offer free disposable items to customers on their own initiative [4]. Citizens are encouraged to purchase and consume recyclable as well as other environment friendly products. 

### 2.2. Measures

To ensure effective implementation of the above new policy, some particular measures have been undertaken by the Shanghai government:

(1) MSW classification publicity. Publicity on MSW classification started in early 2017. Promotional slogans, posters, and videos on MSW classification are presented in newspapers, magazines, televisions, and internet. Brochures are sent to citizens, and promotional activities have been held in the past two years in Shanghai [21,24,26]. Now, the significance of MSW classification has been recognized among citizens of Shanghai [24]. Figure 3 shows brochures, a poster, television video, and a scenario of a promotional activity. 

(2) School education. Teachers in every school are required to educate students on MSW classification, and the students are required to return home and teach their parents what they learned in school [23,27]. Figure 4 presents an educational activity on MSW classification in an elementary school in Changning District, Shanghai. 

(3) Specialist guidance. Volunteers or temporarily hired people, most of whom are enthusiastic retirees, are trained and assigned to some waste collection points to guide citizens to correctly classify waste [21,24,28]. Figure 5 shows a volunteer working at a waste collection point in Huangpu District, Shanghai. 

(4) Incentive methods. Green Accounts, which function as an incentive mechanism, have been widely distributed to Shanghai’s citizens [21,24,29,30]. Figure 6 shows a Green Account card and an automated credit–goods exchange machine of the Green Account in a residential community in Shanghai. Bound with a smartphone, the Green Account records every correct classification of waste and will then give credits, which could be used to exchange for some goods. 

(5) Penalties. Laws and regulations have been introduced for the whole process of throwing, collection, transportation, and disposal of waste. According to the Shanghai Municipal Solid Waste Management Regulation, for example, those who fail to properly classify and/or throw waste will be fined CNY 50 to CNY 200, and waste transportation enterprises that mix the classified waste will be fined CNY 5000 to CNY 50,000 [4]. Figure 7 shows law enforcement officers inspecting at a waste collection point.

## 3. Discussion and Recommendations

It may take a long time to establish a well-developed MSW classification system in Shanghai and the whole of China. Based on the descriptions above, a strengths, weaknesses, opportunities, and threats (SWOT) analysis on the present MSW classification policy in Shanghai was performed, which is tabulated in Table 1. The strengths and opportunities consist of the sufficient infrastructure and technology [20,25], the strong willingness of the government [20], and the increasing willingness of the citizens [24]. There are, however, also various weaknesses and threats consisted in the present policy, and further improvement is needed. 

Existing research shows that leadership and financial support from government are important to MSW management and environmental impact assessments, especially in the initial stage [32,33,34,35,36,37,38]. Currently, most of the waste collection, transportation, and disposal enterprises are owned or subsidized by government companies in Shanghai. The expenditure on human resources, management, and publicity is also majorly supported by the government. The leading and promotional role playing by government now is encouraged to continue in the initial stage of MSW classification policy in Shanghai. It is also noticed that good coordination among different departments of the government is also necessary. However, the burden on management and finance of the government has also increased due to the huge expenditure. The cases of US and Taiwan (China) show that waste collection fees system (WCFS) and private capital and enterprises can be very helpful [10,39]. In the future, WCFS and private capital and enterprises are suggested to be incorporated into MSW classification to establish a sustainable and marketized industry in Shanghai and China nationwide. 

Lu et al. (2006) states that laws and regulations contribute a lot to the success of MSW classification policy of Taiwan (China) [39]. In 2019, Shanghai, ahead of any other cities in China, has promulgated the first local mandatory regulation on MSW classification, and a relatively complete legal system consisting of national and local laws and regulations has been formed [4]. For effective implementation and nationwide promotion, however, the legal system of MSW classification should be further refined and optimized. Moreover, continuous and rigorous inspection and supervision by the government based on the laws and regulations are strongly needed, and those who break the rules should be properly punished. 

People’s willingness, knowledge, and habit on MSW classification play key roles in MSW classification [11,40,41,42,43]. Currently, the vast majority of waste is correctly classified. This, however, is based on a large amount of human and material input. There is a long way ahead for MSW classification to become a habit and a moral principle among people. Further publicity and school education are necessary. It is suggested that the knowledge of MSW classification should be incorporated in textbooks in the near future. More promotional brochures, advertisements, and videos are also a helpful option. 

## 4. Concluding Remarks

This paper reported the new policy on MSW classification in Shanghai, China. The following conclusions are drawn:In recent years, China has changed its development strategy from one based on rapid development to another focusing on environmental protection, and MSW classification is thus reiterated. Shanghai launched China’s first MSW classification policy to adhere to the new national development strategy.Shanghai classifies MSW into four categories: recyclable, hazardous, wet, and dry waste. Various forms of publicity and school education on MSW classification are performed to increase the people’s willingness and public knowledge of MSW classification. Volunteer, incentive, and punitive mechanisms are also adopted to ensure the effective implementation of the new policy.To build a well-developed MSW classification system in Shanghai and the whole of China will require a long period of time. Four suggestions for future development of the MSW classification system in Shanghai and the whole China are presented. First, the leading role of government is encouraged in the initial stage, and good coordination among different departments should be highlighted. Second, WCFS and private capital and enterprises should be involved to promote a sustainable and marketized MSW-relevant industry in the future. Third, laws and regulations should be further refined and optimized, and rigorous and continuous enforcement of the laws and regulations is strongly needed. Fourth, further publicity and school education on MSW classification are also necessary to make MSW classification a habit and a moral principle among citizens.

## Figures and Tables

**Figure 1 ijerph-16-03099-f001:**
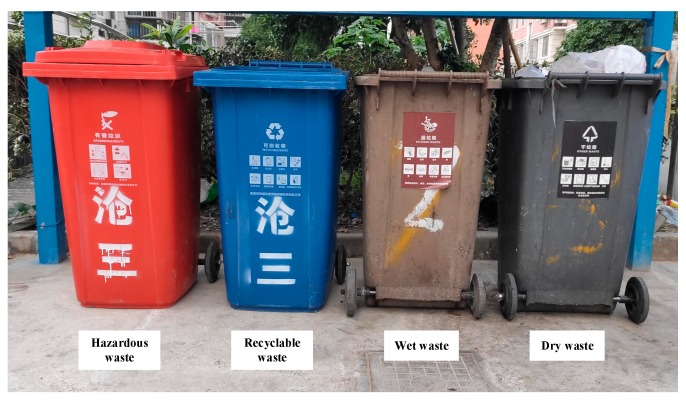
Collection bins of different categories of waste.

**Figure 2 ijerph-16-03099-f002:**
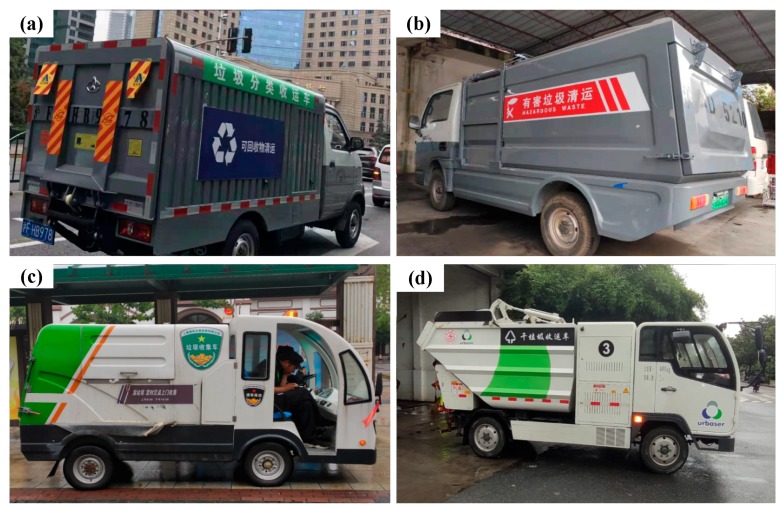
Transportation vehicles of different categories of waste: (**a**) recyclable waste, (**b**) hazardous waste, (**c**) wet waste, (**d**) dry waste.

**Figure 3 ijerph-16-03099-f003:**
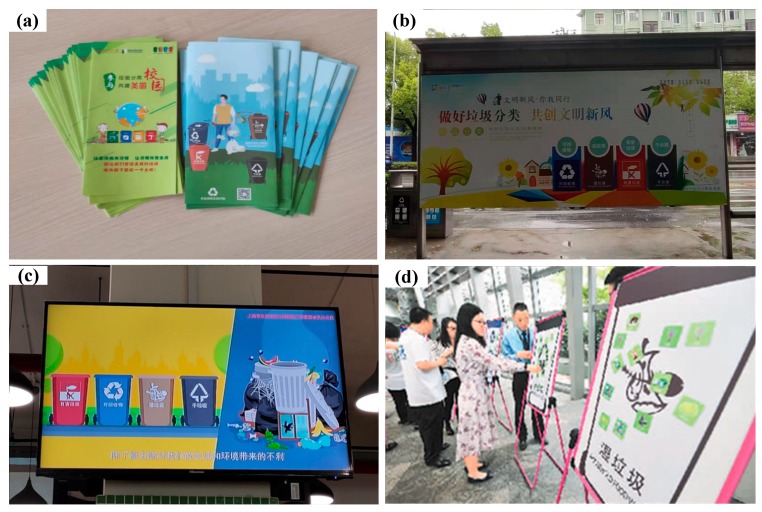
Pictures of municipal solid waste (MSW) classification publicity: (**a**) brochures, (**b**) poster, (**c**) television video, (**d**) promotional activity [26].

**Figure 4 ijerph-16-03099-f004:**
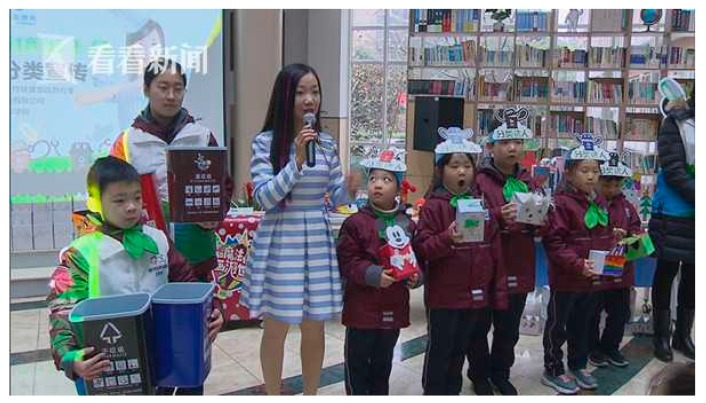
An educational activity on MSW classification in an elementary school in Changning District, Shanghai [27].

**Figure 5 ijerph-16-03099-f005:**
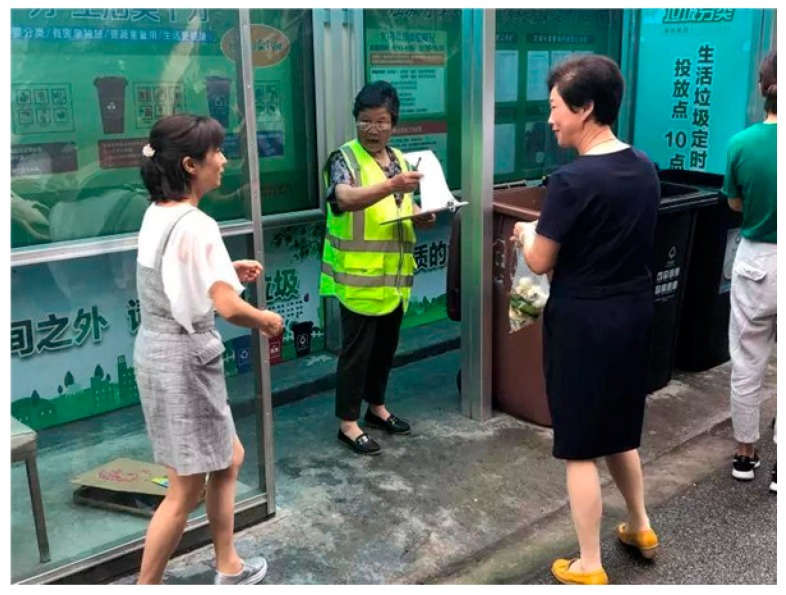
A volunteer working at a waste collection point [28].

**Figure 6 ijerph-16-03099-f006:**
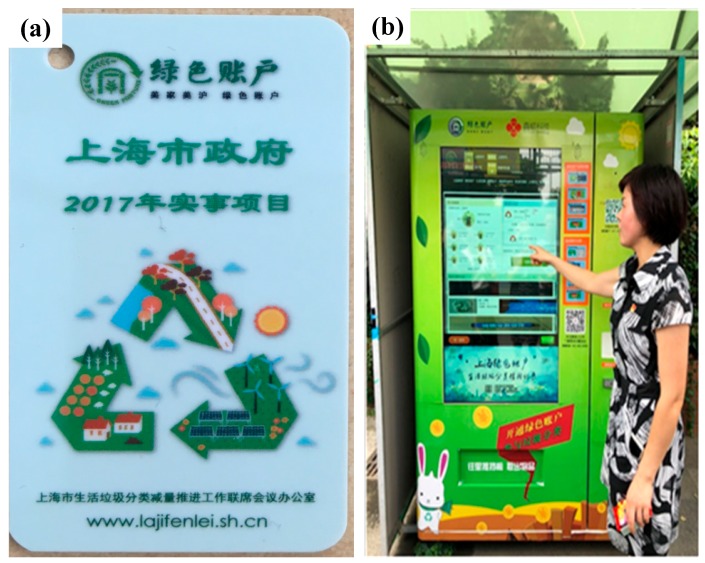
Pictures of Green Account. (**a**) Green Account card, (**b**) automated credit–goods exchange machine [30].

**Figure 7 ijerph-16-03099-f007:**
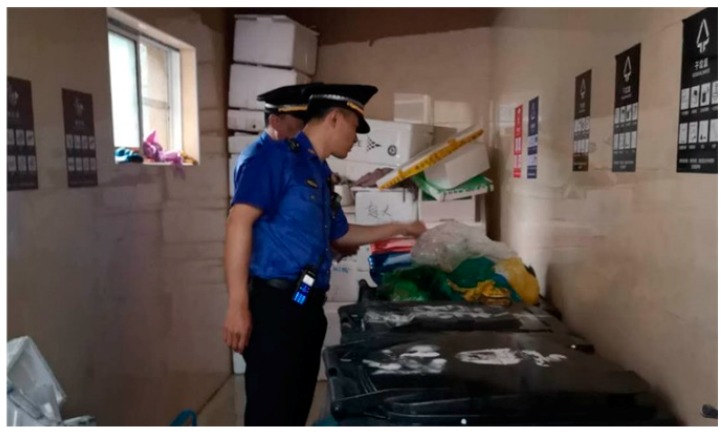
Law enforcement officers inspecting at a waste collection point [31].

**Table 1 ijerph-16-03099-t001:** Strengths, weaknesses, opportunities, and threats (SWOT) analysis on the present MSW classification policy in Shanghai.

Classification	Description
Strengths	1. Sufficient infrastructure and technology.
	2. Strong willingness of the government.
Weaknesses	1. Private capital has not been generally introduced.
	2. Laws and regulations need to be further improved.
	3. Poor coordination among different departments of the government.
Opportunities	1. Willingness of citizens is increasing.
Threats	1. People’s insufficient knowledge of MSW classifications.
	2. Habit has not yet been formed.
Recommendations	1. A leading role and good coordination among different departments of the government are needed.
	2. A marketized industry should be established.
	3. Laws and regulations should be further improved.
	4. More publicity and education are necessary.

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
