# Peer review of "New Policy and Implementation of Municipal Solid Waste Classification in Shanghai, China"

_ijerph, 2019, doi:10.3390/ijerph16173099_

Round 1

Reviewer 1 Report

My impression and assessment of the short communication is as follows.

I find the paper very interesting and bears the hallmarks of novelty and originality. Based on this paper, currently, Shanghai is running a pilot project on waste sorting, of which if successful can be replicated to other Chinese cities. In giving an overview of the research problem or focus, there is a detailed discussion on China's new policy implemented as from 2017. Although the descriptions are adequate, it would help the reader if this policy is referenced so that readers can know where to find this information. 

I am not sure if methods and materials are expected in a short communication, so these can be added if its the journals' requirement. For example, the results are reported as if a longitudinal case study was undertaken. So just provide a brief overview of the the methods.

While the results are interesting enough, its not clear how significant they are because they have not been contrasted nor compared adequately with similar research. To increase and lend more credence to the results, I am asking for a brief discussion on their significance. Are these results  similar or different from findings from elsewhere in the world for projects of this nature, pls address this question?  Lastly, I find the conclusions insightful and properly linked with the study aim and the results generated.

Author Response

Responses to Reviewer 1 Comments

Journal of Environmental Research and Public Health: ijerph-573416

Original Title: New policy and implementation on garbage sorting in Shanghai, China: a brief report

Revised Title: New policy and implementation on municipal solid waste classification in Shanghai, China

Authors: Ming-Hui Zhou, Shui-Long Shen, Ye-Shuang Xu, and An-Nan Zhou

The authors would like to thank the reviewer’s constructive comments, which are very helpful for improving the quality of the paper.

The responses to the reviewer’s comments are detailed below, in which the paragraphs in normal fonts are the comments and the authors’ responses are written in italic fonts. In order to clearly highlight, the revised manuscript is using the "Track Changes" in the resubmitted version.

Reviewer 1 comments:

It would help the reader if this policy is referenced so that readers can know where to find this information

Answer: Thanks for the reviewer’s comment. We have improved the references and cited relevant news (reference 18, 20, 23), government report (reference 16, 17, 25), government regulation (reference 4), and publications (reference 21, 22, 24, 26) in Chapter1 and Chapter 2 to make municipal solid waste (MSW) classification policy of Shanghai clear and well-referenced.

I am not sure if methods and materials are expected in a short communication, so these can be added if it’s the journals' requirement. For example, the results are reported as if a longitudinal case study was undertaken. So just provide a brief overview of the methods.

Answer: Thanks for the reviewer’s comment. We rewrote Chapter 3, and a SWOT (strengths, weaknesses, opportunities, and threats) analysis was conducted in chapter 3. Based on the SWOT analysis, some discussions were added and four recommendations for further improvement of the present policy and measures were also given.

While the results are interesting enough, it’s not clear how significant they are because they have not been contrasted nor compared adequately with similar research. To increase and lend more credence to the results, I am asking for a brief discussion on their significance. Are these results similar or different from findings from elsewhere in the world for projects of this nature, pls address this question?

Answer: Thanks for the reviewer’s comment. In chapter 3, we cited 7 academic papers (reference 27-33) authored by different countries to prove the significance of the role played by government in the MSW classification (see in chapter 3, line 165-166). The cases of Taiwan (China) and US were used to prove that waste collection fee system (WCFS) and private capitals/enterprises are helpful (see in chapter 3, line 173-175). The case of Taiwan (China) was also used to prove the significance of mandatory laws and regulations to the success of the MSW classification (see in chapter 3, line 178-179). Five academic papers (reference 11, 35-38) from different countries were also cited to support the importance of willingness, knowledge, and habit of people in the MSW classification policy (see in chapter 3, line 186-187).

Reviewer 2 Report

Dear Authors, the paper seems appropriate to be published in green newspaper or municipality report however to fit scientifical auditorium it must be significantly improved in such dimensions, it is compulsory!

1) There is no such "garbage", it must be "waste" and classification explained with references what it refers to MSW or C&D waste or Hazardous waste etc. Whether it is defined in China as "garbage" then it should be compared with definitions elsewhere with proper citations

2) Reference list should be significantly improved with authors with various continents as only citing locals is not acceptable for international journal

3) Please add suggestions how this case study can improve the situation in Shanghai, add the SWOT table what benefits and weaknesses You have and analyze otherwise it looks like advertisement

4) What can be learned elsewhere from this case study?

Author Response

Responses to Reviewer 2 Comments

Journal of Environmental Research and Public Health: ijerph-573416

Original Title: New policy and implementation on garbage sorting in Shanghai, China: a brief report

Revised Title: New policy and implementation on municipal solid waste classification in Shanghai, China

Authors: Ming-Hui Zhou, Shui-Long Shen, Ye-Shuang Xu, and An-Nan Zhou

The authors would like to thank the reviewer’s constructive comments, which are very helpful for improving the quality of the paper.

The responses to the reviewer’s comments are detailed below, in which the paragraphs in normal fonts are the comments and the authors’ responses are written in italic fonts. In order to clearly highlight, the revised manuscript is using the "Track Changes" in the resubmitted version.

Reviewer 2 comments:

There is no such "garbage", it must be "waste" and classification explained with references what it refers to MSW or C&D waste or Hazardous waste etc. Whether it is defined in China as "garbage" then it should be compared with definitions elsewhere with proper citations.

Answer: Thanks for the reviewer’s comment. We have changed all “garbage” into “waste” or “municipal solid waste (MSW)”, and “garbage sorting” into “MSW classification” in the paper. Detailed definitions are added in Chapter 1 with some references (see in Chapter1, line 28-line 30).

Reference list should be significantly improved with authors with various continents as only citing locals is not acceptable for international journal.

Answer: Thanks for the reviewer’s comment. As suggested, we improved the references and a total of 12 papers from the international resources that are outside mainland china were cited (references 3, 7, 9, 10, 11, 27, 28, 33, 34, 35, 36, 38).

Please add suggestions how this case study can improve the situation in Shanghai, add the SWOT table what benefits and weaknesses You have and analyze otherwise it looks like advertisement.

Answer: Thanks for the reviewer’s comment. A SWOT analysis on the MSW classification policy of Shanghai has been added in Chapter 3 (see in line 155-161, and Table 1). Based the SWOT analysis, 4 recommendations regarding further improvement of the policy and measures were given (see in Chapter 3, line 165-191; Chapter 4, line205-214). We believe that these recommendations could help to improve the situation in Shanghai.

What can be learned elsewhere from this case study?

Answer: Thanks for the reviewer’s comment. As descripted in chapter 1, china changed its national development strategy from one based on rapid development to anther with focus on environmental protection, the significance of MSW management is reiterated (see in chapter 1, line 41-43). In the future, China would promote MSW classification policy nationwide (see in chapter 1, line 43-44). Shanghai, as one of the most developed cities in China, took the first step of China on WSW classification this year (see in chapter, line 44-48, line 52-60). The policy, measures, and experience of Shanghai might service as a reference and benchmark to other cities in china (see in chapter 1, line 50-52). And, we believe that the 4 recommendations we given in Chapter 3 could also help to improve the MSW classification system of other cities of China (See in Chapter 3, line 155-192; Chapter 4, line 205-214).

Round 2

Reviewer 2 Report

Small grammar style issues